# Identifying key parameters that affect sensitivity of flow tube chemical ionization mass spectrometers

Sneha Aggarwal[1,2], Priyanka Bansal[3], Yuwei Wang[4], Spiro Jorga[3], Gabrielle Macgregor[3], Urs Rohner[3], Thomas Bannan[4], Matthew Salter[1,2,5], Paul Zieger[1,2], Claudia Mohr[6,7], Felipe Lopez–Hilfiker[3]

[1]Department of Environmental Science, Stockholm University, 11419 Stockholm, Sweden
[2]Bolin Centre for Climate Research, Stockholm University, 11419 Stockholm, Sweden
[3]Tofwerk AG, 3645 Thun, Switzerland
[4]Department of Earth and Environmental Science, Centre for Atmospheric Science, University of Manchester, Manchester M13 9PL, United Kingdom
[5]Baltic Sea Centre, Stockholm University, 11419 Stockholm, Sweden
[6] PSI Center for Energy and Environmental Sciences, Paul Scherrer Institute, 5232 Villigen Switzerland
[7]Department of Environmental Systems Science, ETH, 8092 Zürich Switzerland

*Correspondence to*: Claudia Mohr (claudia.mohr@psi.ch), Felipe Lopez–Hilfiker (lopez@tofwerk.com)

**Abstract.** Chemical ionization mass spectrometers are widely used for the detection of trace gases, particularly in the field of atmospheric science. Depending on the analytes of interest, chemical ionization instruments are operated under varying reactor conditions, which can make it difficult to compare instrument performance, even for the same reagent ion chemistry. This variability leads to inconsistent sensitivity distributions, particularly for weakly bound or labile analytes. As a result, determining sensitivity – instrument response per unit analyte concentration – is challenging, even when comparing the same compound detected with the same reagent ion across different studies. To address this issue, we employed multiple Vocus AIM reactors (Tofwerk AG) to systematically identify the critical parameters affecting sensitivity in flow tube chemical ionization mass spectrometers. Controlling these parameters for a given reactor geometry can significantly reduce sensitivity variations across instruments and operators. We demonstrate that sensitivity normalized to reagent ion concentration serves as a fundamental metric for interpreting results from different datasets operating under uniform chemical ionization conditions, such as those within regional networks or other monitoring applications. Calibrating the sensitivity of benzene cations to a group of hydrocarbons, and comparing it to the sensitivity of iodide anions to levoglucosan, a molecule known to react near the collision limit, reveals that it is possible to map kinetic constraints on sensitivity from one ion mode polarity to another, as long as the critical parameters are held constant. Additionally, we show that collision–limited sensitivity relative to the reagent ion is nearly constant across different ionization mechanisms for a given reactor geometry and set of conditions. This consistency enables the determination of the upper limit of sensitivity, even for reagent ions where the specific molecules reacting at the collision limit are unknown. As a result, the use of the voltage scanning approach can be extended to a broader range of reagent ion chemistries. This study highlights how collision–limited sensitivity can enhance our understanding of the relationships between different instruments and simplify calibration requirements across various reagent ion chemistries.

## 1 Introduction

Atmospheric trace gases including volatile organic compounds (VOCs), radical intermediates, inorganic acids, and molecular halogens profoundly affect global oxidative photochemistry, air quality, and climate. Although present in small concentrations, these compounds play a crucial role in tropospheric ozone formation (Atkinson, 2000; Shrivastava et al., 2017), reactive nitrogen species generation (Sillman S., 1999), secondary organic aerosol (SOA) production (Wyche et al., 2014), as well as contribute to atmospheric oxidant cycling (Lelieveld et al., 2016; Yang et al., 2016). Their characterization and quantification remain challenging due to their trace concentrations, diverse chemical compositions, short lifetimes, and dynamic gas–particle phase transitions (Goldstein and Galbally, 2007; Bertram et al., 2009; Mao et al., 2012; Kroll and Seinfeld 2008; Izaacman–VanWertz et al., 2018; Riva et al., 2019).

Chemical ionization mass spectrometry has emerged as a core analytical technique to detect trace gases in the atmosphere. Of particular relevance in the field of atmospheric chemistry are time–of–flight (TOF)–based approaches, which can simultaneously measure hundreds of compounds in real time with detection limits as low as 0.01 parts per trillion by volume (pptv) in the air without sample preparation (Slusher et al., 2004; Ehn et al., 2014; Simon et al., 2016; Riva et al., 2019; Zhang W. et al., 2023; Zhang Y. et al., 2023). Chemical ionization coupled with TOF mass analyzers also exhibits excellent linearity, high time resolution (up to 100 Hz), a wide dynamic range, high sensitivity with sufficient resolving power and mass accuracy to identify compounds in even complex ambient samples (Bertram et al., 2011; Aljawhary et al., 2013; Lee et al., 2014; Breitenlechner et al., 2017; Lee et al., 2018; Priestley et al., 2018; Häkkinen et al., 2023; Alage et al., 2024). Chemical ionization has, therefore addressed a critical need to comprehensively understand the temporal evolution of many key trace gases, even under challenging field conditions (e.g., Huey et al., 2007, Lee et al., 2014; Priestly et al., 2018; Yao et al., 2018; Ye et al., 2021; Bianchi et al., 2022 Huang et al., 2024).

One variant of chemical ionization mass spectrometry is proton transfer reaction (PTR) mass spectrometry, where relatively simple first order reaction kinetics and water cluster distributions govern the overall sensitivity and selectivity of the chemical ionization mechanism. With a narrow distribution of reaction rate constants, quantitative and semi–quantitative conversion of ion signals to concentration is routinely done using a subset of calibrants to validate instrument performance (Riva et al., 2018; Sekimoto et al., 2017; Gouw et al., 2006). Semi–quantitative approaches based on rate constants have become recently more relevant with the widespread introduction of TOF analyzers, which allow detection of hundreds of compounds that are impractical to calibrate individually. While the reaction kinetics and quantification features of PTR–based instruments are practically very useful, the strong electric fields used to control the reagent ion cluster distribution can lead to ionization–induced fragmentation, especially for product ions with labile functional groups, such as –OH, –OOH, and –COOH. Ionization–induced fragmentation complicates the spectral interpretation and limits the amount of chemical information that can be extracted from any PTR mass spectrum (Tani et al., 2003; Aprea et al., 2007; Gueneron et al., 2015;

Gkatzelis et al., 2020; Li et al., 2021; Gkatzelis et al., 2021). As the number of chemicals present in the atmosphere continues to increase due to anthropogenic influence (Gibson et al., 2023 and Wang et al., 2024), the number of possible
fragments and interferences becomes ever more challenging to deconvolve by mass spectrometry alone.

To address the challenges related to ionization–induced fragmentation in PTR and other high energy chemical ionization reaction schemes, flow tube–based chemical ionization systems have gained popularity, particularly for measuring reactive compounds crucial to atmospheric chemistry, which are often particularly prone to fragmentation. Flow tube–based chemical
ionization systems utilize fluid dynamic transport and gentle (thermal) ion–molecule reactions in which abundant reagent ions react with analyte molecules to form product ions. Flow tube reactors generally operate at elevated pressure (50–1000 mbar) to promote adduct formation (ligand switching reactions), charge transfer, or proton transfer/abstraction in the absence of electric fields. Field–free ionization conditions and elevated reaction pressures helps dissipate excess energy from the ion–molecule reactions and preserve the original identity of the analyte ions. A consequence of working at relatively high
pressures and without the presence of electric fields is that controlling the ionization conditions becomes more challenging. Particularly collision conditions and ion energies are less well defined which can lead to shifting sensitivity distributions and/or humidity dependencies. Recent work (Wang et al., 2021; Riva et al., 2024) has provided a framework to suppress humidity effects in flow tube–based chemical ionization reactors, and sensitivity parameterization frameworks for adduct ionization mechanisms are well established (Lopez–Hilfiker et al., 2016a; Zaytsev et al., 2019; He et al., 2024; Song et al.,
2024). Nonetheless, a major challenge remains in understanding and quantifying all the key parameters that define sensitivity distributions so they can be adequately controlled. This includes both temporal variability and differences between instruments of the same type operated differently, which together contribute to the net observed sensitivity variability.

Sensitivity ($S_i$) in a CIMS is defined as the signal ($\psi_{N,i}$), normalized to a standard value of 1 million reagent ions per
90 second measured at the detector, per unit analyte concentration ($C_i$), as shown in Eq. (1). It depends on the chemical properties of the analyte, such as polarity, structure, proton affinity, ionization energy, and available functional groups, especially in adduct–based mechanisms. Instrument parameters – such as temperature, pressure, reaction time, water content in the reaction volume, and voltage of the transfer ion optics also play a significant role. All these factors can be summarized by two main components governing sensitivity – the net formation rate of product ions in the reactor cell, and the
95 transmission efficiency of these ions in their intact form to the detector, as shown in Eq. (2) (Lopez–Hilfiker et al., 2016a).

$$S_i \;=\; \frac{\psi_{N,i}}{c_i}, \tag{1}$$

$$\psi_{N,i} = \left[\int k_f([X][i]dt) \,\times\, T^i\left(\frac{m}{q}, B_i\right) dt\right] \times \left(\frac{1}{X} \times 10^6\right), \tag{2}$$

$$= [Product\ ion\ formation\ rate\ \times Transmission\ efficiency]\ \times Normalized\ reagent\ ion\ intensity$$

In Eq. (2), $\psi_{N,i}$ is the normalized signal of analyte $i$ in reaction time $t$ at a product ion formation rate of $k_f$ given $[X]$ is the reagent ion concentration and $[i]$ is the analyte concentration. $T^i$ refers to the ion–specific transmission efficiency, which depends on the mass–to–charge ratio $\left(\frac{m}{q}\right)$, and the product–ion binding/bond energy ($B_i$) relative to the electric field strength of the ion optics. Analyte signals in flow tube reactors are routinely normalized to 1 million ion counts per second of reagent ion as measured at the detector. This practice helps to eliminate variations in instrument response due to changes in reagent ion source intensity, or detector gain (Huey et al., 2007), essentially using the reagent ion signal as an internal standard. With the introduction of wider detected mass ranges and brighter ion sources, it is important to note that reagent ion normalization is only valid in mass spectral regions where the relative ion transmission remains approximately constant (Fig. S1) and where detector saturation does not occur. Significant variations in transmission efficiency could otherwise introduce a considerable mass–dependent bias in normalized or relative responses. Additionally, detector saturation would skew the observed sensitivity normalized to reagent ions, resulting in artificially high values.

To accurately quantify the normalized response of a chemical ionization mass spectrometer (CIMS), some calibrations are needed to determine analyte concentrations by measuring the net reaction rates and transfer efficiency in the reactor and analyzer. In practice, it is often impossible or at the very least impractical to calibrate every compound observed. Calibration efforts may be further complicated due to the lack of available standards, the reactive nature of compounds, or due to the sheer number of compounds that are routinely detected and should be quantified. Iyer et al. (2016) showed that the reaction rate constant for most of the commonly studied multifunctional molecules varies by less than a factor of two when using iodide as the reagent ion, consistent with the relative narrow distributions of PTR rate constants (Sekimoto et al., 2017). However, variations in ionization pressures, reactor geometries, reaction times, and instrument tuning can lead to significantly varying reaction and transmission conditions, ultimately resulting in relatively wide sensitivity distributions. Consequently, calibration results even for the same molecule detected with the same ion chemistry can vary significantly complicating the community's effort to improve quantification capabilities.

From first principles, if all critical parameters could be precisely controlled for a set of reagent ions and reactor geometries, the observed sensitivities and sensitivity distributions between compounds across different instruments should become much narrower than what is typically reported, and should actually become identical within experimental uncertainty. If such a state could be realized, it would in principle facilitate not only the transfer of calibration factors across time, but also between instruments with relatively low error. This would represent a major advantage for distributed datasets collected by instruments of the same type operating under the same general chemical ionization conditions in regional networks or other research and monitoring disciplines.

In an effort to achieve more consistent sensitivity distributions, our study identifies and quantitatively evaluates the key factors influencing signal response for a given ion chemistry, including, manufacturing intolerances, flow rates entering the reactor, effective reaction temperatures, voltage gradients, and reaction times. To this end we compare individual instrument responses to different classes of compounds across 39 Tofwerk AG (Thun, Switzerland) mass spectrometers configured with Vocus AIM flow tube reactors. We assess the statistical variability in normalized instrument performance through

quantitative comparison across multiple instruments to gain insight into what parameters govern sensitivity, and therefore control accuracy and precision. We also introduce a simplified framework for the determination of the collision limit using multiple reagent ions, which can be used in combination with existing voltage scanning approaches to estimate instrument sensitivity towards individual ion–molecule adducts without requiring direct calibration (Lopez–Hilfiker et al., 2016a). Furthermore, we demonstrate how sensitivity normalized to the reagent ion signal can be used as a fundamental property

across different reagent ions, which aids bulk quantification efforts and simplifies the exploration of new reagent ions. By addressing these objectives, we provide a framework to unify sensitivity across different individual instruments, thus enabling more efficient, comparable, and quantitatively robust atmospheric trace gas measurements crucial for enhancing our understanding of complex atmospheric processes and their impacts on global climate and air quality. This work improves the consistency and reliability of flow tube CIMS in general, which is relevant also beyond the field of atmospheric sciences.

**2 Experimental methods**

    In this study, we examined 39 different mass spectrometers of various models, all configured with the same Vocus AIM reactor, to evaluate the statistical distributions of sensitivities towards select compounds. A schematic of the Vocus AIM instrument interface, including the relevant ion optics, chamber pressures, and flow rates, is shown in Fig. 1 and discussed in detail in Riva et. al. (2024). Each instrument was operated under standardized conditions: 50 mbar reactor pressure, 50 °C

reactor temperature, and consistent collision energies by using the same electric fields in the first stages of the mass spectrometer. While the absolute sensitivities on any given instrument may be optimized by further adjustment of the interface voltages, for the purposes of this study these voltages were held constant in regions where ion–neutral collisions occur (Fig. 1, grey shading). The models and specifics of each instrument type along with detailed ion optic voltages at standard conditions are summarized in Table S1.

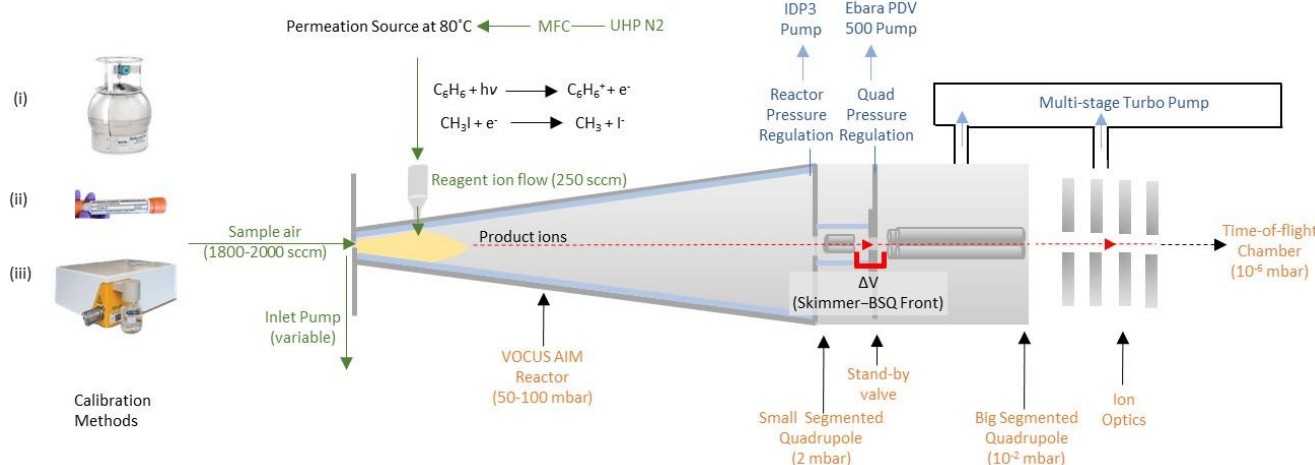

**Figure 1. A schematic of the High–Resolution Time–of–flight Chemical Ionization Mass Spectrometer (HR–TOF CIMS) with a Vocus AIM flow tube reactor. The sample air enters the reactor through a 0.5–mm pinhole, where the analyte molecules are ionized at 50 mbar. Subsequently, the air is sub–sampled through a 1–mm pinhole to the quadrupole stage of the differentially pressured interfaces. We used three different methods for introducing standard calibration gases to the reactor: (i) multi–component certified gas cylinders, (ii) permeation tubes, and (iii) liquid calibration system (LCS).**

Generally, we tuned the instruments so that all static interface voltages in the first two differentially pumped regions (ion–molecule reactor (IMR) and transfer quadrupole) are set to 0 V (i.e., ground). A radio–frequency (RF)–only quadrupole is used to focus ions in the radial direction (~100 peak–to–peak voltage ($V_{p-p}$) 3.2 MHz), while the gas jet transfers the ions axially towards a skimmer at the entrance of the second segmented quadrupole. Subsequently, the ions undergo a limited number of energetic collisions with neutral molecules cooling the ion population before entering a lens stack and orthogonally pulsed for mass analysis in the TOF mass analyzer. Energetic collisions in the first stages of the interface are limited by low electric field strengths (E/N, Td) in an effort to promote and preserve ion–molecule adducts. It is important to note that these collisions can be manipulated to control observed cluster distributions and transmitted adduct formation, but we define conditions that generally promote the formation of ion–molecule clusters rather than that of the bare ionized analyte. We focus on how key ion optical elements and reactor conditions influence both absolute sensitivity and relative sensitivity (reproducibility independent of accuracy) distributions, as these are critical for quantitative analysis.

## 2.1 Vocus AIM reactor and reagent ion generation

The Vocus AIM reactor is a commercially available flow tube–based IMR, described in detail elsewhere (Riva et al., 2024). It operates at medium pressure (50–100 mbar) and is conical in shape with conductive walls to reduce the loss and memory of reactive trace compounds. It uses field–free ionization conditions with ion molecule reactions following optimized fluid dynamic flow to ensure efficient mixing, while also minimizing turbulence and wall losses. Part of this optimization is the

180 reduction of the reagent–ion laden nitrogen stream to 250 sccm, an order of magnitude lower than conventional designs, which helps to minimize turbulence in the first stages of the reactor, where sample and reagent ion flows meet. The reactor is entirely made of conductive polytetrafluoroethylene (PTFE) and is temperature–controlled, which results in fast response times even for classically sticky molecules (e.g., acids, amines). Reagent ions are generated in abundance by vacuum ultraviolet (VUV) lamp (Ji et al., 2020; Breitenlechner et al., 2022; Riva et al., 2024) in a small, dedicated chamber, that

isolates the VUV radiation from the reaction cell, injecting the resulting ions directly into the sample flow. Here, we focus primarily on two reagent ions, benzene cations and iodide anions. These ions are formed simultaneously by illuminating an ultra–high purity (UHP) nitrogen stream, carrying trace amounts of methyl iodide and benzene (released from a permeation tube held at 80 °C; ~1:100 by volume). In the ionization process, vapours released from the permeation tube directly absorb light in the 117–124 nm band to form benzene cations and a free thermal electron. The thermal electron is then scavenged by

methyl iodide, resulting in rapid thermal dissociation to form an iodide anion and a neutral methyl radical. These ions are then promptly injected into the reactor with the small carrier flow of nitrogen at an angle of 45 degrees as shown in Fig. 1.

We chose benzene cations and iodide adducts because they are among the most commonly used ion chemistries in flow tube reactors. These ions provide complimentary detection capabilities for both hydrocarbons and oxygenated organic and

195 inorganic compounds with relatively little overlap. This allows us to make conclusions across a larger range of chemical species than any single ion would support. Moreover, they are generated by a single ion source, making them an ideal choice for evaluating reproducibility of ion chemistry in this study. This approach reduces uncertainty related to transfer efficiencies, ion yields, and mixing dynamics when comparing the performance of different reagent ions across instruments or different reagent ions on the same instrument. While these ion chemistries are individually well described in previous

work (Huey et al., 1995; McNeil et al., 2007; Lee et al., 2014; Kim et al., 2016; Lavi et al., 2018; Leibrock and Huey, 2000), a systematic analysis of their statistical reproducibility and key properties affecting sensitivity variability remains critically needed.

### 2.1.1 Iodide adduct chemical ionization

Iodide anions are widely used for the measurement of reactive and oxidized organics, as well as a suite of inorganic

molecules via adduct formation. A generalized adduct–forming reaction of a given analyte (R) with iodide can be represented by one of the following reaction pathways (Huey et al., 1995; Slusher et al., 2004; Kercher et al., 2009; Lee et al., 2014, Lopez–Hilfiker et al., 2016a; Robinson et al., 2022; Breitenlechner et al., 2022).

$$I^- + R \; \rightarrow \; I^-.R \tag{R1}$$

$$I^-.(H_2O)_n + R \; \rightarrow \; I^-.R + n(H_2O) \tag{R2}$$

$$I^- + RH \; \rightarrow \; R^- + HI \tag{R3}$$

$$I^-.R + E^* \rightarrow I^- + R \tag{R4}$$

Third bodies (including water vapour; R2) play an important role in net adduct formation either by carrying away excess energy from the collision (stabilization) or by limiting the availability of iodide through competition (Lee et al., 2014; Riva et al., 2024). Iodide ions generally form weak adducts with most organic molecules with binding energies on the order of 15–30 kcal mol$^{-1}$ (Iyer et al., 2016). Such weak interaction energies limit fragmentation of analyte molecules as the intermolecular bond energies are much higher than the energy of the adduct complex. In relatively rare thermodynamically favourable cases, iodide ions may also abstract a proton (H$^+$) from the analyte molecule (R3), for example, sulfuric acid, resulting in a charged anion (Lee et al., 2014). Bowers et al. (2023) showed that perfluoroalkyl sulfonic acids and polyfluoroalkyl phosphoric acid diesters are also detected as deprotonated anions because of their low proton transfer enthalpies. While this mechanism is generally limited for organic compounds, some peaks without an iodide attached are routinely observed but are not broadly used for analysis. The transfer of weakly bound iodide adduct complexes through mass spectrometer interfaces requires careful control of the total energy ($E^*$), which includes both thermal and kinetic components. Excessive energy from poorly optimized ion optics can cause these adducts to dissociate, reversing reaction R3 to produce neutral analytes and iodide anions (R4). This energy control during both reaction and transfer stages is therefore crucial for maximizing sensitivity in iodide adduct–based detection methods.

### 2.1.2 Benzene cation chemical ionization

Benzene cations are generally sensitive towards hydrocarbons such as VOCs, their first–generation oxidation products, and other lightly oxygenated VOCs, as well as ammonia. While many of these classes could also be detected with a PTR–based approach, in many cases fragmentation of larger analyte molecules can limit their quantitative measurement (Kim et al., 2009). The ionization mechanism for benzene and benzene cluster cations has been extensively discussed elsewhere (Kim et al., 2016; Lavi et al., 2018; Horning et al., 1973). In summary, an analyte is either detected as a cation via charge transfer (R5), or as an adduct via ligand switching (R6). It may also be detected as a protonated cation when the gas–phase basicity of the analyte is greater than that of the phenyl radical (R7).

$$(C_6H_6)^+ + R \rightarrow (C_6H_6) + R^+ \tag{R5}$$
$$(C_6H_6)_2^+ + R \rightarrow R^+(C_6H_6) + C_6H_6 \tag{R6}$$
$$(C_6H_6)^+ + R \rightarrow C_6H_6 \cdot + RH^+ \tag{R7}$$

The ionization pathway for a given analyte molecule in benzene mode mainly depends on its ionization energy (IE). A molecule with IE lower than that of the benzene dimer (8.69 eV) is expected to be detected as a cation, while a molecule with a higher IE is expected to be detected as an adduct (Kim et al., 2016; Lavi et al., 2018). Most hydrocarbons are detected as cations (R5) with intramolecular bond strengths stronger than the excess energy imparted during ionization or by the electric field of the ion optics during transfer to the detector. Some compounds, such as ammonia (IE=10.07 eV) and isoprene (IE=8.86 eV), which cannot be efficiently ionized, can be detected as adducts (R6) with relatively weak binding

energies, similar to iodide adducts. With respect to humidity dependence, benzene cations cannot directly ionize water molecules due to the higher IE of water (12.6 eV). However, under high humidity, benzene cations may partially hydrate in the reactor, particularly at elevated reactor pressures, which may result in a humidity–dependent sensitivity (Ibrahim et al., 2005; Miyazaki et al., 2004). As a result, humidity suppression or control systems are often implemented in atmospheric sampling or where humidity conditions change over time.

### 2.1.3 Additional reagent ion chemistries used in this study

In this study, we primarily use iodide and benzene as reagent ions to determine the key factors controlling sensitivity and evaluate if normalized sensitivity is a fundamental metric under uniform chemical ionization conditions. However, we have also used other reagent ion chemistries – bromide and nitrate anions, and protonated acetone and ethanol dimers – to demonstrate applicability of normalized sensitivity to field conditions as well as to evaluate if collision-limited sensitivity is independent of reagent ion for a given flow tube geometry and set of conditions. Here we summarize the basic properties of these different reagent ions as implemented here, noting that they are well described elsewhere in the literature (Sanchez et al., 2016; Albrecht et al., 2019; Simon et al., 2016; Kürten et al., 2014; Ehn et al., 2014; Prabhakar and Vairamani, 1997; Dong et al., 2022; Yao et al., 2016; Yu and Lee 2012).

Briefly, bromide and nitrate reagent ions are similar to iodide in a sense that they also generally detect polar and oxidized organic and inorganic compounds via adduct formation (Lawler et al., 2011; Jokinen et al., 2012). They are generated by a VUV ion source in a similar fashion as iodide ions, but with different additives as scavenger of the thermal electrons. Bromide ions are formed by introducing trace di-bromomethane in the presence of benzene or acetone and nitrate ions by instead introducing trace nitric acid in the presence of benzene. Protonated ethanol dimers and acetone dimers are generated by introducing the respective vapor into the VUV ion source. Bromide ions are especially useful to measure a wide range of iodine containing species (Wang et al., 2021), as well as chlorine and chlorinated species (Lawler et al., 2011), hydroperoxy radical (Sanchez et al., 2016), and sulfuric acid (Rissanen et al., 2019). Nitrate ions are very selective and have been routinely used to measure highly functionalized species, such as oxygenated volatile organic compounds (OVOCs) and highly oxidized molecules (HOMs) (Garmash et al., 2024, Alage et al., 2024). At relatively high neutral concentrations of nitric acid in an atmospheric pressure chemical ionization using nitrate ions, most of the reagent ions are clustered with at least one nitric acid molecule. At low neutral nitric acid concentrations, the cluster distribution shifts towards more bare $NO_3^-$ ions, which decreases the selectivity of the ionization scheme. This is because at lower neutral nitric acid concentrations, weakly interacting analytes do not have to compete as much with neutral nitric acid to form an adduct with $NO_3^-$ (Hyttinen et al., 2015). In benzene cation mode, at extreme differences in concentration, similar effects have been reported (Lavi et al., 2018). Protonated acetone and ethanol dimers are used primarily for solvent–type molecules. They undergo proton transfer or adduct formation chemical reactions with basic analytes, such as, gaseous atmospheric ammonia

and organic amines (Nowak et al., 2006; Ye and Lee, 2012; Dong et al., 2022) for which the proton affinity of the analyte is favourable or when the reagent analyte binding energy is sufficiently high to result in a stable adduct.

## 2.2 Calibration methods and standards

To quantitatively calibrate and systematically measure the side–by–side relative and absolute responses of several instruments, we employed three complementary calibration methods: (1) multi–component certified gas cylinders for quantitative calibration of hydrocarbons, (2) permeation tubes for high–volatility compounds, and (3) a liquid calibration system (LCS) for lower–volatility compounds. For correlation analysis between individual instruments, we introduced temporally variable concentrations into a common inlet and evaluated the correlation in normalized response between all connected instruments.

### 2.2.1 Gas–phase standards

Standard gas mixtures of known concentrations were generated by certified multi–component standards provided by Apel Riemer Environmental, Miami, United States. Initial cylinder concentrations of 1–10 parts per million (ppm) were dynamically diluted by mass flow controllers (Bronkhorst Model: TOF–101) to generate standard concentrations of 0–10 parts per billion by volume (ppbv) for calibration purposes. The dynamically diluted gas was directed to the calibration and zero port of the Vocus AIM reactor, which introduces the gas in a total flow of 2000 sccm to overflow directly upstream of the IMR entrance pinhole. We primarily used a 13–component PTR calibration cylinder in nitrogen of which 6 compounds were detected by benzene cations. While we used these compounds to evaluate the sensitivity of all instrument models, multiple individually produced cylinders of the same nominal concentration were used in the calibrations reported here. The nominal cylinder accuracy is $\pm 5$ % and the propagated error in the dynamic dilution is approximately $\pm 10$ %.

### 2.2.2 Permeation–based standards

Commercially available permeation tubes were used to introduce standard concentrations into the inlet of the instruments. For most of the experiments, the permeation tubes were used only for correlation analysis between collocated instruments and dynamically diluted into a common sample flow to mitigate differences in absolute emission rates of the permeation tubes over time. In the scenario where permeation tubes were used for quantitative calibration, they were placed in a temperature–controlled perfluoroalkoxy (PFA) oven and purged by a small nitrogen flow to equilibrate. The resulting vapours were diluted into a larger carrier flow resulting in a final concentration of less than 10 ppbv before being introduced into the inlet. We used VICI Metronics Dynacal permeation devices for formic acid (PD–2850–UR), nitric acid (PD–0160–UR), and ammonia (140–693–0140–U50) with permeation rates of 30 ng min$^{-1}$ $\pm$ 50 % at 30 °C, 74 ng min$^{-1}$ $\pm$ 25 % at 40 °C, and 50 ng min$^{-1}$ $\pm$ 25 % at 50 °C, respectively.

### 2.2.3 Liquid calibration system

For compounds with relatively low volatility or reactive functional groups that cannot be produced using conventional gas standards, permeation tubes, or similar methods, we generated a standard concentration in situ using an LCS from Tofwerk AG, Switzerland (Riva et al., 2024; Wu et al., 2023). The LCS can generate gas standards from liquids of known concentration by quantitative evaporation and dilution. Aqueous solutions with ~20 micro molar concentrations were dispensed through a liquid flow controller into a heated evaporation chamber. At the entrance of the chamber, a nebulization system generates droplets which are digested (and evaporated) during transit through the heated oven (Song et al., 2024; Xu et al., 2022). The volatility range accessible depends on the thermal stability and vapour pressure of the compounds present in the solution. We find empirically that volatilities down to that of levoglucosan equilibrate in a reasonable timescale (minutes). However, the addition of acid and other sticky moieties becomes more problematic from a time response and quantitative transfer perspective (Khare et al., 2022; Coggon et al., 2018). We estimate the uncertainty of the gas standard produced by the LCS as +10 % to −25 % by propagating the error from the liquid flow meter, gas mass flow controller, stock solution preparation, subsequent dilution, and the potential losses of the calibrant inside the evaporation chamber. We note that due to the elevated gas temperature of the standard exiting the LCS, care needs to be taken when calibrating weakly bound adducts that are thermally unstable for quantitatively accurate results.

## 3. Results and discussion

### 3.1 Key factors controlling sensitivity

The sensitivity of chemical ionization instruments is influenced by two main factors: (1) the formation rate of product ions, governed by collision frequency and the available energy in the reactor, and (2) the transmission efficiency of these product ions to the detector. While the number of collisions and available energy primarily depends on temperature and pressure conditions integrated across the reaction time in the reactor, the transmission efficiency depends on the product ion stability relative to the electric field strength of the ion optics as well as the mass–to–charge ratio of ions relative to the quadrupole bandpass window. In the following sub–sections, we evaluate these key factors affecting the two terms that define sensitivity.

### 3.1.1 Effect of reactor pressure

In a flow–tube IMR, the reaction time is set by the mass flow rate through the reaction volume, and therefore the reactor pressure and flow rate play a dominant role in net instrument sensitivity. The sensitivity increases approximately quadratically as a function of pressure due to changes in collision frequency and a proportionally equal increase in reaction time (residence time) at constant mass flow rate through the reactor. The response of the normalized sensitivity across the relevant pressure range of the Vocus AIM is shown in Fig. 2(a), along with a model based on the changes in collision

frequency and reaction time anchored at 30 mbar reactor pressure empirically. We observe a factor of ~4 increase in sensitivity for a doubling of reactor pressure (from 30–60 mbar) with all other parameters remaining the same. The increased

sensitivity at higher absolute pressures is likely also due to the flow branching effects at the reactor exit. At this point, the gas flow splits between two paths: one to the next differentially pumped region of the mass spectrometer, and the other to the IMR vacuum pump. At higher pressures, a larger fraction of the IMR flow preferentially enters the next differentially pumped region rather than the fore pump, further enhancing ion transmission. While this effect is measurable as the deviation between the model and the observations, it remains a relatively minor effect compared to the collision frequency

and residence time effects of changing pressure. Additional secondary effects like changes in the water cluster distribution or ratio of neutral to charged reagent ions could also affect the reagent cluster distributions and, in some cases, affect sensitivity distributions, primarily for weakly bound adducts or compounds whose ionization energy is closer to that of the reagent ion.

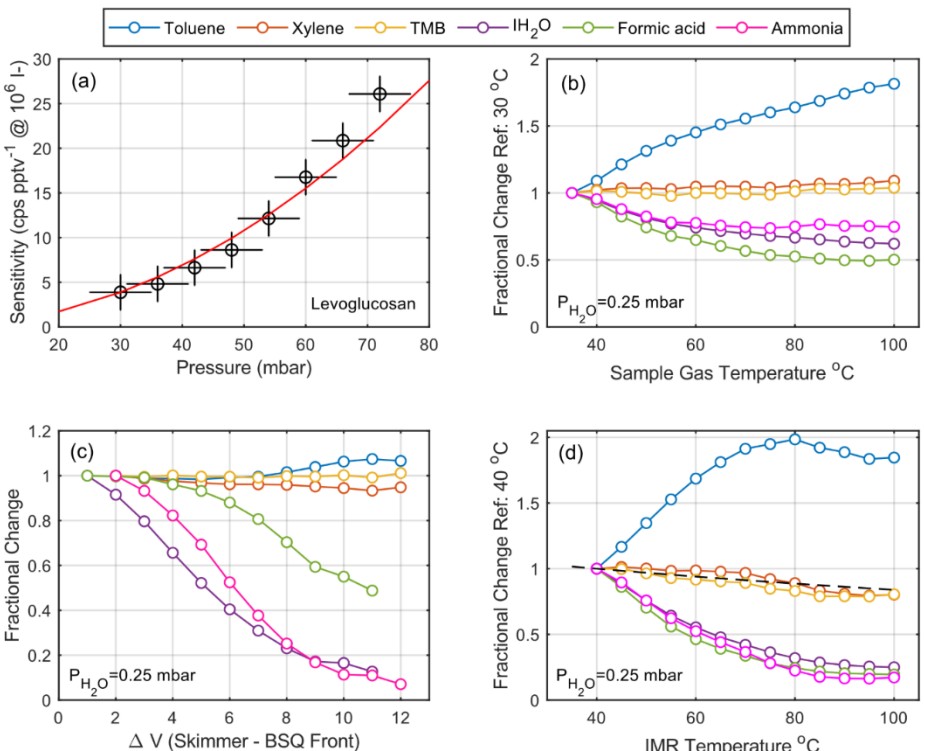

**Figure 2. The dependence of sensitivity on key instrumental and sample parameters. (a) Collision–limited sensitivity as a function of reactor pressure as measured by levoglucosan with iodide anions. The combined effects of collision frequency and reaction time anchored at 30 mbar reactor pressure is modelled as a red line. (b) The sensitivity dependence as a function of sample gas temperature varied from 25 to 100 °C. (c) The sensitivity dependence of iodide adducts as well as hydrocarbon cations, including the benzene cation adduct with ammonia as a function of the voltage gradient in the transfer optics after the reaction cell. (d) The**

**sensitivity dependence on the temperature of ion-molecule reactor (IMR) at a fixed sample temperature of 25 °C. The black dashed line represents the calculated change in sensitivity by reduction of residence time in the IMR as a function of temperature. A change in sample gas temperature (from 25–100 °C) corresponds to an equivalent change in IMR temperature of ~20 °C.**

### 3.1.2 Effect of sample gas temperature and reactor temperature

There are no electric fields present in a flow–tube IMR to control the mean energy of the reagent ions, so the only
parameters available to directly adjust the energy during ionization are the reactor temperature and the incoming gas
temperature. The most common manipulation of sample temperature is the use of heated inlets with the goal of improving
the transmission of low–volatility material down long inlets. While this effort is often futile (Stark et al., 2017; Lopez–
Hilfiker et al., 2016b; Zhao et al., 2024) due to secondary effects involving partitioning, thermal degradation, and reaction
with inlet walls, it can also significantly impact sensitivity distributions. Figure 2(b) shows the dependence of the net
reaction efficiency (sensitivity) on the incoming sample gas temperature. In this experiment, the sample gas temperature
directly before the IMR entrance was modulated from 25 to 100 °C by transit through a 5–cm heated metal tube using a
resistive heater. The temperature of the gas was recorded by a miniature PT–100 temperature sensor mounted directly
upstream of the IMR entry orifice. The sample gas flow carried a constant concentration of hydrocarbons, ammonia, and
formic acid to evaluate the changes in reaction kinetics and cluster distributions during the heating and cooling of the sample
stream. We chose these compounds because of the span of reaction pathways (adduct formation, charge transfer) as well as
sufficiently high volatility to not to have significant wall effects during the thermal manipulation.

Except for toluene, the sensitivity of other hydrocarbons, such as trimethylbenzene (TMB) and xylene, remained essentially
constant despite the significant change in the sample gas temperature. This is expected, as these compounds are primarily
detected as charge transfer products, and their ionization pathway is not significantly influenced by the added thermal
energy. However, toluene which ionizes via a very similar mechanism exhibits a very different behaviour, showing an abrupt
increase in sensitivity with increasing temperature. We attribute this change in toluene response to its strong water vapour
dependence (Fig. S2) and relatively high ionization energy. This observation is consistent with the addition of thermal
energy shifting the reactant cluster distribution in the IMR. We hypothesize that at higher sample temperatures, the reagent
cluster distribution in the reactor shifts towards drier (bare) reagent ions due to the increased thermal energy and therefore
less in source clustering of benzene with water vapor occurs which hinders the ionization of toluene by competition. This
effective declustering of water bound to benzene in the reactor effectively counters the effect of water vapour in the reaction
between benzene and toluene by removing water vapor as a competitor.

Molecules detected as adducts (ammonia, water vapour, and formic acid) decrease systematically with increasing sample
temperature. This is because the binding energy of ion–molecule adducts is generally low and therefore the increased
thermal energy from the heated gas leads to ion-molecule dissociation in the IMR. As a result, weaker adducts like the
iodide–water cluster with a binding energy of ∼10 kcal mol$^{-1}$ (Iyer et al., 2016) dissociate more rapidly than adducts with
higher binding energy. Increasing sample gas temperature therefore induces a binding energy dependent shift in sensitivities
for any adduct–based ionization mechanism. This effect should be taken into account when using heated inlets, such as the

Filter Inlet for Gases and Aerosols (FIGAERO; Lopez–Hilfiker et al., 2014; Thornton et al., 2020; Yang et al., 2021) and temperature–programmed thermal desorption systems (Smith and Rathbone, 2008; Winkler et al., 2012; Li et al., 2021). It is also important to take this effect into account when the temperature of the sample air changes significantly during ambient measurements, such as when measurements are taken at different altitudes during an aircraft campaign or at different locations during mobile measurements.

The reactor manifold temperature also plays a critical role in defining the ion energies, and therefore the sensitivity distribution. Reactor walls are often moderately heated with the goal of controlling the reaction conditions, eliminating diurnal temperature fluctuations in both laboratory and field settings, and to improve the time response for lower volatility compounds. The consequence of heating the reactor is that it also adds internal energy to the analytes, which can lead to dissociation of weakly bound ion–molecule adducts and in extreme cases to thermal decomposition of analytes. These effects have been well documented in the literature (Horning et al., 1973; Huey and Lovejoy, 1996; Huey et al., 1998; Huey, 2007; Lee et al., 2014; Sanchez et al., 2016; Robinson et al., 2022). Figure 2(d) shows the overall dependence of sensitivity on the reactor temperature at a fixed sample gas temperature of 25 °C. While the same general patterns are observed, perturbations in reactor temperature have a significantly greater impact on the cluster distribution and reaction kinetics for all compounds, with the effect being especially pronounced for adducts. This is consistent with the walls of the reactor playing a more important role as a heat source than the sample gas, which has limited heat capacity. Heating the reactor walls also affects the net gas density in the reactor, but this is a relatively minor effect that slightly reduces (~20 %) the reaction time (Fig. 2(d), dashed line). This effect broadly explains the response of TMB and xylene, as these compounds have no additional thermal dependencies and are therefore only affected by the change in reaction time.

In a relative sense, a change in sample gas temperature from 25 °C to ~100 °C is roughly equivalent to a change of about 20 °C in the reactor temperature. This indicates that reactor walls are a far more efficient means of stabilizing reactant ion distributions against ambient temperature changes, achieving stabilization at lower absolute temperatures and thereby likely reducing issues from analyte thermal decomposition or dissociation. In practice, operating the reactor and inlet at the lowest feasible temperature supports adduct formation while having minimal impact on hydrocarbon detection. However, the controlled temperature should be sufficiently high enough to prevent adsorption or smearing of the analytes of the interest, and remains above typical environmental variations s in the vicinity of the reactor and inlet.

### 3.1.3 Effect of voltage gradient in the ion optics

The voltage gradient in the ion optics is another crucial parameter that controls the net observed sensitivity as the product ions move through the reactor and the differentially pumped interface of the mass spectrometer. In general, declustering and activation of product ions must be minimized in the first stages of the ion optical interface. Calculating the declustered fraction of an adduct from first principles can be difficult due to the small distances between electrodes, time–varying

electric fields, and jet expansion between pressure stages (Olenius et al., 2013). Consequently, empirical approaches have

425 been developed to examine the stability of product ions. One such approach is described by Lopez–Hilfiker et. al. (2016a), where a voltage scanning procedure is proposed to experimentally determine the binding energies of detected ion adducts. As the stability of adducts depends critically on the binding energy between the reagent ion and the analyte molecule, relative to the energy added during analysis (Iyer et al., 2016, Lopez–Hilfiker et al., 2016a), the voltage gradient on the way to the detector can be programmatically adjusted such that binding energies can be determined. We used voltage scanning

approach to investigate the relative influence of voltage gradients on sensitivity in the first stages of the differentially pumped interface of our mass spectrometers. The potential difference between the skimmer and the first element of the big segmented quadrupole (BSQ) held at $\sim 10^{-2}$ mbar was adjusted in steps of 1 V and the corresponding changes in detected ion intensity were recorded. We only scanned this region of the instrument because all other interface regions either have no voltage gradient, or the mean free path is larger than the distance between neighbouring electrodes. Figure 2(c) shows the

result of this scan for selected analytes detected using benzene cations and iodide adducts. As expected, the recorded signal for weak adducts decreases rapidly as a function of increasing voltage gradient, as the collision energy imparted to them exceeds their binding energy with the reagent ion. Meanwhile, hydrocarbon sensitivity remains unchanged within the experimental error, as the energy imparted is much less than the bond energies of the molecules therefore no fragmentation is observed.

In summary, the key factors controlling instrument sensitivity with a flow–tube IMR include reactor pressure, effective ion temperature, and voltage gradients applied in the ion optics (Fig. 3). Increasing reactor pressure generally increases sensitivity due to an increase in collision frequency and reaction time but can have some penalizing side effects. Higher pressure promotes the formation of higher–order water clusters at a given humidity, which exacerbates water vapour effects

and accelerates reagent ion titration, thereby reducing the upper limit of the linear range. Additionally, higher pressure may facilitate enough time for secondary ion chemistry, thereby complicating ionization mechanisms (Zhang and Zhang, 2021; Breitenlechner et al., 2022; Robinson et al., 2022). Effective ion temperature is determined by a balance of thermal energy (from reactor and sample temperatures) and the energy imparted on product ions during transit through the mass spectrometer. Operating at lower effective temperatures has several advantages related to adduct formation and preservation

but maintaining the IMR at 40–50 °C is a practical compromise for preserving weak adducts and stabilizing against most ambient temperature changes. Buffering the incoming sample air temperature can further dampen diurnally driven changes in sensitivity and limit biased measurements of diurnal profiles. This is especially important for weakly bound adducts or compound which have strong humidity dependencies. Among the ion–optical parameters, the voltage difference between the skimmer and the entrance of the second transfer quadrupole is the most critical. This is the final stage in the interface where

significant collisions occur, and the pressure is low enough that small changes in absolute voltages matter. In the further downstream vacuum stages, ion molecule collisions no longer play a significant declustering or activation role. Approximately equivalent net activation can be estimated using proportional changes to different parameters. For example,

using formic acid, we can equate a voltage difference of ~5 V in the entrance to the second quadrupole region to a binding energy of ~23 kcal mol$^{-1}$, which is approximately equivalent to amount of added energy as a reactor temperature change of ~20 °C based on the change in the formic acid sensitivity. While the factors mentioned above are crucial for understanding the sensitivity of an individual chemical ionization instrument, they also define the parameters which should be well controlled to ensure consistent performance across multiple instruments.

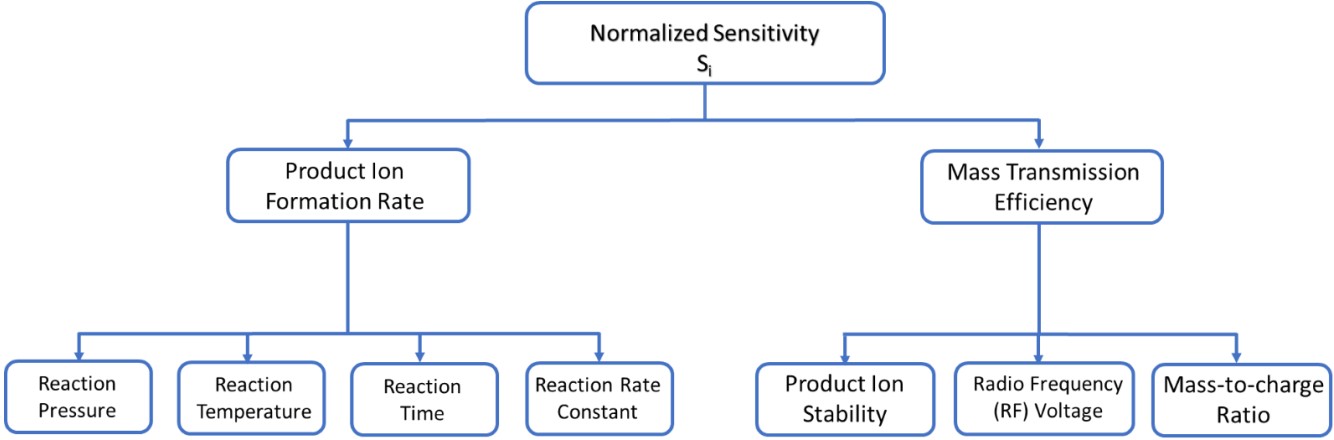

Figure 3. A branching diagram showing the key parameters that influence the instrument sensitivity ($S_i$), derived from the signal normalized to the reagent ion in a flow tube chemical ionization reactor.

## 3.2 Reproducibility of normalized sensitivity

It is common practice to normalize analyte signal to the observed reagent ion across time on a flow tube–based chemical ionization instrument. This helps to correct for any fluctuations in reagent ion source intensity and compensate for changes in detector gain, as well as minor variations in the previously mentioned key parameters. As a result, normalization in regions with constant relative transmission efficiency (for different BSQ resonant frequencies and as a function of mass–to–charge ratio, Fig. S1) stabilizes small changes in instrument sensitivity across time in most cases. While reagent ion normalization is widely used on a single instrument, demonstrating that this normalized performance is a fundamental property across different instruments has proven to be significantly more challenging. This is in part because operational conditions and reactor geometries may vary significantly between different groups (e.g., IMR pressure 20-950 mbar) which makes it difficult to obtain a statistically significant number of instruments operated similarly needed to draw reliable conclusions.

If however, the relevant parameters of Eq. (2) are held constant and are well controlled across multiple instruments and reactors, then one would expect that their normalized sensitivities should be comparable or even identical. To practically evaluate the feasibility of this, we systematically calibrated a number of instruments (39) and assessed their normalized response to compounds sensitive to the parameters outlined in Sect. 3.1. We interpret deviations in the observed normalized

sensitivity across instruments as a direct reflection of instrument variability, which encompasses both potential variations or biases from control systems and assembly or production tolerances. These biases may stem from factors such as offsets in

pressure measurement, inconsistent flow rates, poor mixing, inadequate temperature control, or manufacturing tolerances (e.g., orifice diameter, electrode spacing, etc.). Each of these factors can impact the net normalized sensitivity. Recognizing that calibration bias could also significantly influence instrument comparisons, we assess individual calibrations (Sect. 3.2.1) and conduct co–sampled correlation analyses under both laboratory (Sect. 3.2.2) and field conditions (Sect. 3.2.3) to evaluate reproducibility of instruments across different scenarios.

**3.2.1 Statistical analysis of normalized sensitivity**

We calibrated 39 instruments with benzene cations and iodide anions to determine the variability in the normalized sensitivity. These instruments included 5 Vocus B, 2 Vocus B2, 3 Vocus B4, 22 Vocus 2R, 5 Vocus S, and 2 Vocus Scout units, with resolving power ranging from 1,200 to >10,000 Th/Th. While all instruments are based on the same interface and ion optics, the key differences relate to the polarity switching timescale. The Vocus B, B2, and B4 units are bipolar mass

spectrometers with a unique flight path in the TOF region for positive and negative ions while sharing a common interface. They are capable of fast polarity switching within 50 ms by rapid switching of the interface voltages, which are all less than 165 V and can therefore reliably be switched quickly. A typical measurement cycle corresponds to each of up to 4 reagent ions being used for 0.5 seconds, in a 2–second measurement loop (Fig. S7). The other models evaluated require roughly 10 minutes for polarity switching as low voltages and high voltages need to be changed. Fast changes of high voltages

otherwise can damage electronics or cause discharges in the TOF analyzer or detector assembly. We used a hydrocarbon gas mixture to quantify the performance of benzene cations to toluene, xylene, α–pinene and TMB, and an LCS to calibrate the response of iodide anions to levoglucosan. As flow is a critical parameter in determining both the dilution and the reaction time in the reactor (Fig. S3), the corresponding change in reaction time was taken into account for instruments with more than one VUV ion source (+250 sccm per additional source). We also standardized the inlet flow rates which have some

variability from the manufacturing process of pinhole plates to a standard flow rate of 1600 sccm by compensating the measured flow rate deviations to the standard flow rate with a direct impact on the reaction time (see Fig. S3).

The calibration results from the 39 instruments are summarized in Fig. 4, as a box and whisker plot that groups normalized instrument responses by analyte compound. This demonstrates remarkably consistent normalized sensitivities across the

510 different compounds. The net variability is on the order of ±15 % for toluene and xylene, with slightly higher deviations observed for α–pinene and TMB. We attribute the larger variability of these compounds to their higher relative stickiness, which leads to attenuation and long equilibration times in the calibration lines and flow controllers of the instruments. This is also consistent with an asymmetric error bar biased towards lower normalized sensitivity. As the calibration gas purging time, laboratory temperature and cylinder temperature were not systematically controlled, variability may also arise from

515 differences in the calibrant gases equilibration times in the gas lines, flow controllers, and regulators. The calibration of

levoglucosan with iodide adducts exhibits slightly higher variability and a slightly lower mean. This is likely due to challenges inherent in calibrating levoglucosan, a multifunctional semi–volatile compound that is solid at room temperature, as well as possible variability in the solution preparation. While we cannot completely exclude transfer and sampling losses between the LCS evaporation chamber and the reactor (Khare et al., 2022; Coggon et al., 2018), given the time response to
changes in concentration occurs on the timescale of minutes we estimate this effect is on the order of ~10% or less.

We selected levoglucosan as the calibrant for the negative ion mode because previous studies have reported that it reacts near the collision limit within the bounds of experimental uncertainty (Lopez-Hilfiker et al., 2016). However, as part of our statistical comparison and investigation into the apparent bias between the collision limit as determined by hydrocarbon
calibrations relative to that of levoglucosan (Fig. 4), we determined that levoglucosan is likely to be systematically lower than the collision limit by about ~20 % with iodide anions (Fig. S6). This is in contrast to other adduct-forming anions like bromide ions (see Sect. 3.4). This helps to explain the systematic shift of levoglucosan relative to the hydrocarbon sensitivities in Fig. 4.

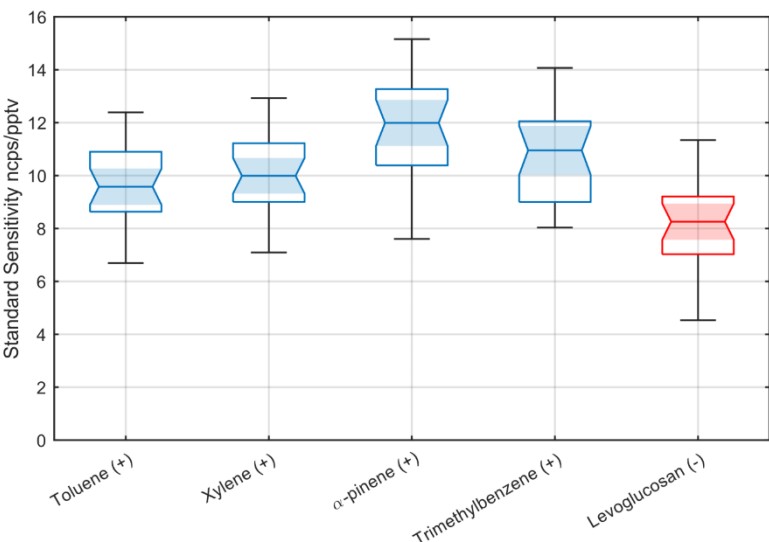

**Figure 4. Statistical analysis of 39 Vocus AIM instruments calibrated in benzene cation mode (shown in blue) by a series of hydrocarbons under dry conditions, which ionize at approximately the collision–limited sensitivity of 10 cps pptv$^{-1}$ at a total reagent ion current of 1 million ions s$^{-1}$, using a certified gas standard. (ncps: normalized counts per second) For comparison, iodide anions (shown in red) calibrated with levoglucosan show a very similar sensitivity. Levoglucosan is known to ionize with iodide anions near the collision limit. The absolute value of the sensitivity is**
**consistent with a global normalized collision–limited sensitivity independent of reagent ion. Each box shows the interquartile range (IQR), i.e., 25$^{th}$ to 75$^{th}$ percentiles of the data points. The horizontal line in each box indicates the median value, while the notch around the median illustrates a 95% confidence interval. The shading serves to**

**enhance the visual emphasis on the confidence interval of the median. The whiskers extending from the edges of the box indicate variability outside the middle 50% of the data.**

Overall, the individual calibration variability, as defined by the average response and the spread for each compound across instruments, remains remarkably small. This is particularly noteworthy given the best–case scenario cannot exceed the stated accuracy of our gas calibration standards from Apel Riemer Environmental, which is ±5 % for hydrocarbons or the propagated total uncertainty of ±10%. Even for levoglucosan, which is inherently more challenging to calibrate, the

545 estimated uncertainty of +10 to –20 % is consistent with the calibrations from the hydrocarbon measurements. These results suggest that most key parameters affecting normalized sensitivity are well–constrained and controlled. However, some outliers are observed as indicated by the whiskers. Given the otherwise tight distribution, we attribute these outliers to potential calibration errors, which may include issues such as leaks in the mixing valves or gas connections or insufficient equilibration time after starting the calibrant flows (less than two hours).

**3.2.2 Normalized variability when co–sampling under laboratory conditions**

To further investigate the variability in normalized response across multiple instruments in the absence of calibration biases outlined above, we calibrated a subset of 4 instruments (all Vocus B models) using a common sample line. We systematically introduced formic acid, nitric acid, chlorine, ethylene glycol, acetone, and hydrocarbons into a 5–meter 3/8" PFA inlet flushed with 10,000 sccm of clean air, from which all instruments were co-sampling, and compared their

normalized response. We used a lag correlation analysis to synchronize the time response of each instrument, but the lag was often within 1–2 sample points (2–4 seconds).

Figure 5 shows the result of these simultaneous additions performed using a common sample line. The correlation of each individual instrument against the mean of all responses is represented in the form of a bar plot. Using a common inlet line removes the potential bias due to individual calibration errors and reduces the total variability in response (sensitivity) across

instruments to approximately ±10 % for most of the volatile compounds. Notably, nitric acid exhibits a larger net variability than the other more volatile compounds. This is likely due to sampling biases and partitioning across the 5–meter unheated inlet across which each instrument sampled at intervals of ~1 meter. In contrast, compounds like formic acid and chlorine, which are detected using the same ion chemistry, exhibit significantly lower net variability as their attenuation in the sample tubing is minimal. When considering all compounds together, the median variation between instruments is approximately of

±15 %. This level of variation is excellent between individual instruments, as it is not substantially higher than that of a single individually calibrated instrument. This demonstrates that normalized response indeed can be reproduced across a larger compound distribution and between different individual instruments under relatively controlled laboratory conditions.

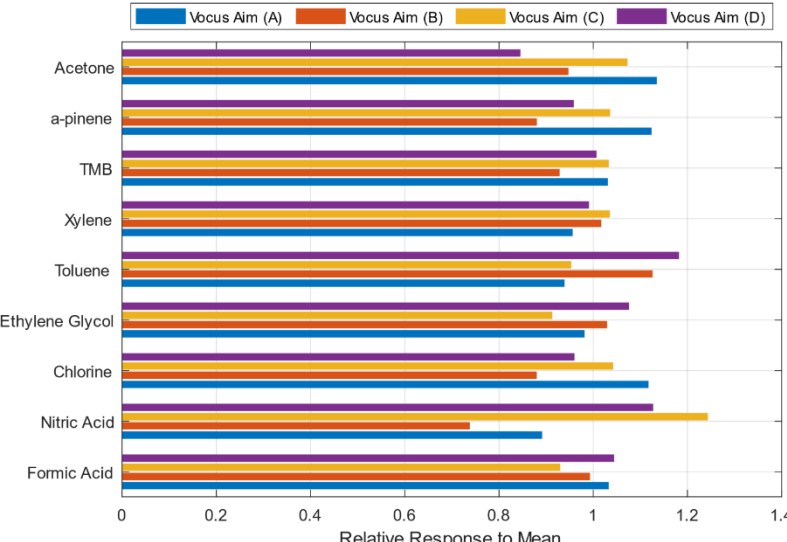

**Figure 5. Correlation of co–sampling instruments in a laboratory setting and calibration variability. Total variability is reduced as the instruments are simultaneously calibrated, minimizing the variability associated with multiple individual calibrations, which have a larger propagated error.**

### 3.2.3 Normalized variability when co–sampling under field conditions

To evaluate the applicability of normalized instrument response under more challenging field conditions, we conducted ambient air sampling with four Vocus B instruments at the Tofwerk facility in Thun, Switzerland, a small city situated in a valley near a large lake. The sampling location was on the fourth floor of the Tofwerk AG building in a quiet neighbourhood. Sample air was drawn through an unheated PFA inlet at ~20,000 sccm, positioned 1 m from the building's exterior wall. The sampling line extended 6 m through an indoor room, where each instrument sampled sequentially. All mass spectrometers were operated at identical reactor conditions and interface voltages, consistent with previously described experiments. We used Vocus B instruments because these instruments can switch quickly between different reagent ions, which allowed us to quasi–simultaneously compare compounds detected by benzene, iodide, and protonated acetone reagent ions in the sample air.

Figure 6 and Fig. S4 show the results in the form of the time series of 4 selected analyte molecules measured in the sample air, namely nitric acid, nitrous acid (HONO), methyl ethyl ketone (MEK), and TMB. We chose them because they are all well isolated peaks and exhibit a large temporal variability during the measurement period. All the instruments effectively captured the temporal variations in ambient analyte concentrations, with each of them showing very consistent responses. Moreover, the inter–instrumental variability in terms of reagent normalized response is generally within ±15 % for all compounds except nitric acid. Nitric acid demonstrates low net variability at the beginning of the measurement period; however, the variability increases notably between March 19 and 21. During this period, elevated laboratory temperatures

due to direct sun likely impacted the sampling efficiency and inlet artifacts of the different instruments, with the effects dependent on their positions along the sample line. Since other compounds measured by the same ion chemistry during the measurement period do not show this deviation, consistent with findings from the laboratory–based correlation analysis, we conclude that the greater variability observed in nitric acid arises from the inherent challenges in its comparison, rather than differences in instruments' overall response. The combined results from individual calibrations, laboratory correlations, and

ambient intercomparisons demonstrate that normalization can effectively standardize responses across instruments when key sensitivity parameters are held constant. Under standardized operating conditions, this allows sensitivity mapping between similarly configured instruments with estimated uncertainties of ±10–20 %.

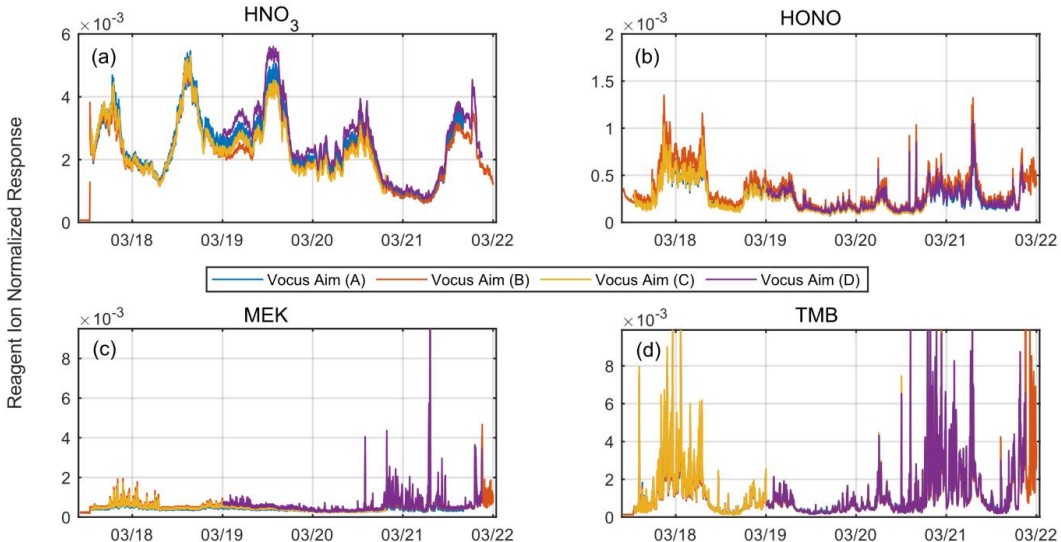

**Figure 6. Simultaneous measurements of ambient air using 4 different Vocus AIM instruments, (a) with nitric acid (HNO₃), (b)**
**nitrous acid (HONO), (c) methyl ethyl ketone (MEK), and (d) Trimethylbenzene (TMB), as examples. Each instrument used the same interface voltages and was connected to a common high flow sampling line. Each instrument switched between 3 different measurement modes (iodide, acetone, and benzene) every 2 seconds. The timeseries of individual ions normalized to the respective reagent ion demonstrates excellent temporal agreement across the measurement period.**

## 3.3 A simplified method for constraining the collision limit

Calibrating instrument responses for all detected compounds using a CIMS is challenging, error prone, and time consuming for all the reasons discussed in Sect. 1. As a potential solution to this problem, Lopez–Hilfiker et al. (2016a), proposed a voltage scanning approach combined with the knowledge of collision limit to estimate relative sensitivities for compounds lacking calibration standards or for bulk quantification of observed ion signals. The combination of the collision limited sensitivity provides a theoretical/experimental upper limit for sensitivity, while voltage scanning assesses the stability of

product ions by empirically evaluating binding energies. This empirical approach allows estimation of relative sensitivities, i.e., determining how closely any given compound approaches the collision–limited sensitivity and is of particular utility for compounds for which standards are unavailable or cannot be quantitatively introduced.

Extending this approach to other reactor geometries and reagent ions necessitates knowledge of molecules that react at the collision limit with the given reagent ion to define the maximum sensitivity from which all other sensitives are derived. However, accurately identifying and characterizing such molecules is challenging as they are not well defined or easily measured. For example, calibration work in a reduced pressure flow tube IMR by Huey et al. (1995) has shown that $N_2O_5$ reacts with iodide at the collision limit. However, the reaction of iodide anions with $N_2O_5$ proceeds via two channels ($IN_2O_5^-$, $NO_3^-$) depending on the electric field conditions in the transfer optics (Kercher et al., 2009). This dependence, combined with variations in mass transmission efficiency makes it complicated to determine the collision–limited sensitivity. Moreover, it is challenging to generate quantitative in situ source of $N_2O_5$. These challenges ultimately make collision–limited sensitivity determinations using $N_2O_5$ unlikely to be widely adopted for experimental determinations of the collision limit of iodide adduct chemical ionization. Another molecule known to react near the collision limit with iodide is levoglucosan (Lopez–Hilfiker et al., 2016a). However, it is relatively sticky and readily partitions to the walls of the sampling lines making efficient delivery to the reactor or sample inlets a challenge. As a result, few independent collision–limited sensitivity values are reported in the literature, despite a large number of reactor geometries and reaction conditions in regular use.

To overcome the challenges of reporting the collision–limited sensitivity, we demonstrate a comparatively simple approach by utilizing two reagent ions from a single ion source on a single reactor. This ensures that the reactor geometry, flow rates, reaction time, and ion–molecule mixing are all identical. As a result, the only free parameters left in Eq. (2) are the absolute number of reagent ions available for reaction and the relative ion transmission efficiency. Therefore, we expect collision–limited sensitivity to be the same for both the positive and negative ions generated from the same bipolar source when normalized to reagent ion concentration in regions of uniform transmission efficiency (Fig. S1). As the resulting ion currents recorded at the detector only depend on the relative transmission of the instrument and interface, sensitivities can easily be standardized by normalizing to the reagent ion current as a measure of relevant reagent ion transmission.

It follows that it should be possible to calibrate an instrument in either of the two ion modes to determine the collision limit at constant reaction conditions and then derive the normalized maximum sensitivity from one ion mode to another of opposite polarity. Using benzene mode, we calibrated a series of hydrocarbons with ionization potentials lower than benzene, which typically react at the collision limit (Leibrock and Huey, 2000). Under dry conditions, we observed a consistent sensitivity of $10\pm2$ cps ppt$^{-1}$ at a reagent ion current of 1 million ions per second measured at the detector (Fig. 4). We compared results with levoglucosan sensitivity in iodide mode, previously shown to react near the collision limit (Lopez–Hilfiker et al., 2016a) and to first order, the sensitivity agreed within experimental uncertainty between the two ion chemistries, supporting our hypothesis.

We chose to demonstrate this approach using benzene cations and iodide adducts because it addresses several challenges calibrating collision–limited sensitivity with iodide adducts. First, hydrocarbon–based calibration standards are readily available with good analytical accuracy ($\pm$5–10 %) and are relatively simple to handle, and second, most of the hydrocarbons undergo charge transfer with benzene to form cations and therefore are transferred through the ion optical interface of the mass spectrometer without significant perturbation. Therefore, in an instrument configured with both iodide anions and benzene cations, quantification of net changes in reaction conditions in the IMR is much more easily tracked using benzene cations than iodide adducts from a collision–limited sensitivity perspective.

### 3.4 Towards generalized CIMS sensitivity and application of collision–limited sensitivity

So far, we have focused primarily on the dependence of benzene cations and iodide adducts; however, there are many other ion chemistries in regular use, which have the same calibration challenges presented in Sect. 1. A key conclusion from Sect. 3.3 is that ions operated at the same reactor conditions should exhibit the same collision–limited sensitivity when normalized to the reagent ions. To further test this, we generated multiple reagent ions in the Vocus AIM reactor and calibrated with compounds expected to ionize near the collision limit. We calibrated benzene cations using xylene, TMB, and alpha pinene while iodide anions with levoglucosan, perfluoropentanoic acid (PFPeA), and perfluorobutanoic acid (PFBA), as shown in Table S2. In addition to these, we calibrated bromide anions with chlorine and levoglucosan, comparing their response to levoglucosan detected with iodide ions. We calibrated protonated acetone and ethanol dimers with a series of amines and propylene glycol monomethyl ether acetate (PGMEA) that have much higher proton affinities than protonated acetone or ethanol and therefore can be assumed to react at near the collision limit. For nitrate anions, we used perfluorohexanoic acid (PFHxA), the most strongly bound PFAS from commercially available PFAS calibration standards. The results for all these ion chemistries are shown in Fig. 7 and Table S2. In general, all calibrations fall within a remarkably narrow range, supporting the hypothesis that collision–limited sensitivity is indeed independent of reagent ion selection for a given reactor geometry and consistent reaction conditions (Fig. 3).

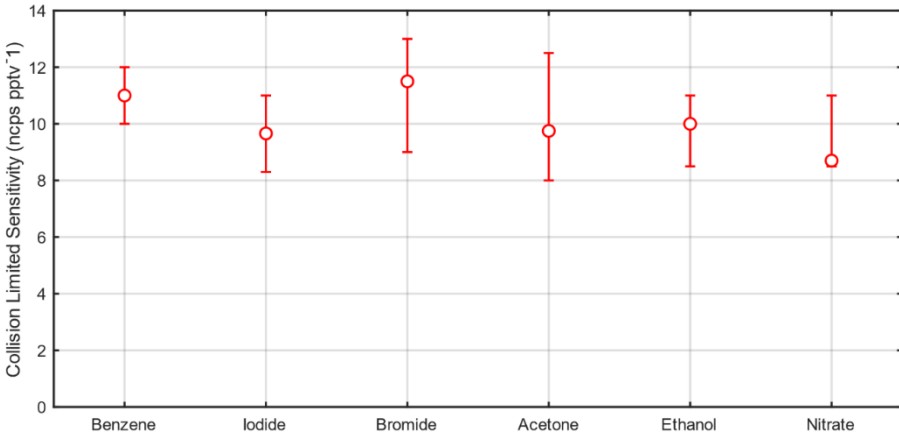

**Figure 7. Experimentally determined sensitivities for different reagent ions operated under standard conditions. For included compound selection see Table S2. Error bars reflect the calibration bias in the case of a single compound – or the range of calibrations treated as near the collision limit including experimental uncertainty. (ncps: normalized counts per second)**

We observe that some compounds are detected at more than one peak under our standard operating conditions of the transfer

ion optics. For example, dimethylaniline is detected partially as a protonated ion and partially as an adduct with acetone (Fig. S5). This degree of clustering can be controlled by the collision energy imparted during transfer to the detector after exiting the reactor. The overall reaction rate in the reactor as presented in Eq. (2) should follow the collision limit in the case of an amine with very high proton affinity; however, the degree of clustering or protonation is entirely dependent on the transfer conditions in the ion optical interface – a factor that first order kinetics does not account for. Rather, the sum of the detection

channels, whether a protonated amine or a higher order cluster with the reagent ion (e.g., acetone or ethanol), should be consistent with the collision limit, as the distribution of charges to different detection channels (peaks) must conserve the number of ionized analyte molecules per reagent ion collision in the reactor. Scanning the declustering energy as shown in Fig. S5 demonstrates this concept, where the sum sensitivity for ethanolamine and dimethylaniline remains essentially constant as a function of the declustering voltage while the fractional distribution of peaks changes. One could select the

optimal cluster distribution based on this type of scan to promote or prevent adduct formation in an otherwise mixed detection pathway.

Given the relatively narrow range of collision–limited measurements observed for each of the reagent ions, despite their vastly different ionization mechanisms, we conclude that collision–limited sensitivities relative to the reagent ion are nearly

690 constant for a given flow tube geometry and operation conditions (temperature, pressure, flow rates, and electric field). This suggests that the upper limit of sensitivity can be determined for a broader range of reagent ion chemistries, even for those where the specific molecules reacting at the collision limit are unknown. For example, this approach can be applied to nitrate adducts, even when operated at different pressures or in cases where the extrapolation of sulfuric acid calibration to other compounds becomes limiting (Ehn et al., 2014; Kirkby et al., 2016; Alage et al., 2024). As a result, the primary challenge for

most adduct–based ion chemistries shifts from determining parameterized sensitivity to quantifying how close a given molecule is to the collision limit, or in cases where multiple peaks are detected per analyte, understanding how the signal is fractionally distributed.

Our measurements demonstrate a collision–limited sensitivity of $10 \pm 2$ cps ppt$^{-1}$ at a reagent ion current of 1 million cps broadly for the Vocus AIM reactor at standard conditions (50 mbar reaction pressure, 10 ms reaction time), independent of the reagent ion used. Despite the somewhat lower normalized sensitivities of the AIM reactor, absolute sensitivities are approximately similar to literature values as VUV–based ion sources routinely generate ion currents of 3–6 million ions per second (Lopez–Hilfiker et al., 2016a; Iyer et al., 2016; Gramlich et al., 2024; Riva et al., 2024). At absolute sensitivities of 30–60 cps pptv$^{-1}$, and background count rates of <1–10 cps, 3–sigma detection limits based on counting statistics can be calculated to be in the range of 0.05–0.15 pptv consistent with the measured detection limit for compounds that have no significant chemical background and are primarily limited by ion counting statistics. In scenarios where the sensitivity still remains limiting, the collision–limited sensitivity can be increased by raising the reactor pressure, as shown in Fig. 2(a), at the expense of linear range.

## 4. Conclusions

We have identified and characterized the key parameters that control sensitivity of a CIMS equipped with a Vocus AIM reactor. These parameters include reactor pressure, effective ion temperature (regulated by the balance between the reactor and sample temperatures), and voltage gradient in the interface ion optics. An increase in reactor pressure generally results in an increase in sensitivity due to increase in reaction time and collision frequency within the confines of a workable linear range. As flow tube reactors explicitly do not have electric fields, the only source of additional kinetic energy is heat from the sample flow and the reactor walls. Elevated effective ionization temperatures can dissociate weakly bound product ions even at relatively low absolute temperatures and – if elevated high enough – can also lead to thermal fragmentation of some functional groups. The effects of sample temperature are most important to consider when using heated inlets, especially for weakly bound adducts or when effective sample gas temperatures are cycled. Maintaining constant collision conditions in the interface where the mean free path is shorter than the distance between ion optical elements is also critical to maintaining a constant sensitivity distribution in any instrument. We note that in this manuscript, we focus on instrument sensitivity, however, the measurement setup, including sampling line material, residence time, and relative humidity and temperature changes of the sample, can influence the net sensitivity function and should be carefully assessed for the desired measurement goals (Neuman et al, 1999; Kürten et al., 2012; Lee et al., 2014; Breitenlechner et al., 2017; Krechmer et al., 2018; Riva et al., 2019; Li et al., 2019).

We demonstrated that controlling each of these dependencies allows normalized sensitivity to become a fundamental metric across different reagent ions and different instruments independent of their model. Our unique results obtained by individually calibrating 39 independent instruments provide a foundation for mapping instrument performance between conditions, but – equally important – between instruments, provided the relevant parameters are well controlled and held constant. The reagent–normalized response remains robust even under field conditions, maintaining consistency across different analyte concentrations and ambient conditions.

We also demonstrate a simplified approach for mapping normalized collision–limited sensitivity between reagent ion modes of opposite polarity under constant reactor geometry and operating conditions. Our findings demonstrate that collision–limited sensitivity remains consistent across different reagent ion chemistries (benzene cations, iodide anions, protonated acetone dimers, protonated ethanol dimers, bromide anions, and nitrate anions) within experimental uncertainty. This independence from reagent ion chemistry provides an improved framework for predicting sensitivities in reduced pressure flow tube reactors. The empirical measurement procedure we demonstrate requires minimal effort and reduces experimental error, establishing a foundation for broader understanding of flow tube sensitivities. When combined with voltage–scanning approaches, this framework can improve sensitivity predictions without requiring detailed knowledge of molecular structure or other physical properties. These advances in understanding flow tube sensitivity distributions enable better synchronization of sensitivity across instruments, provide a framework for sensitivity distributions simplifying calibration requirements, and improve measurement comparability between different research groups. This represents a significant step toward enhancing measurement consistency and reliability, while providing a foundation for future sensitivity parameterization work.

**Data Availability:**

Data for the figures presented in this manuscript are available by written request to lopez@tofwerk.com.

**Author contributions:**

FL, CM, and SA came up with the research idea and designed the study. SA, PB, SJ, GM, UR, YW, TB, and FL conducted the experiments, while SA, PB, SJ, and FL analyzed the data. SA and FL drafted the manuscript, and all co–authors reviewed and provided critical feedback, contributing to its clarity and coherence. Additionally, all co–authors made substantial contributions to discussions and provided valuable feedback.

**Competing interests:**

F.L–H, P.B., S.J., G.M., and U.R. are employees of Tofwerk AG who commercialized the AIM reactor coupled to time–of–flight mass spectrometers.

**Acknowledgements:**

We would like to express our gratitude to Inga Rosenberg, Lars Machler, Irena Miladinova, Patrick Read, and Sabrina
Waldburger, Jonas Fahrni, and Stefan Stubenhofer for their careful work calibrating the numerous individual CIMS
instruments with a Vocus AIM reactor at Tofwerk AG in Thun, Switzerland.

**Financial support:** This work was supported by the Swedish Research Council (Vetenskapsrådet; grant number: 2020–
05025).

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
