# Peer review of "Identifying key parameters that affect sensitivity of flow tube chemical ionization mass spectrometers"

_EGUsphere, 2025_

## Author Response (AR1)

**Response to Reviewers' Comments – Manuscript ID [egusphere-2025-696]**

*Dear Editor,*

*We thank the reviewers for taking the time to review our manuscript and for their valuable feedback. We have carefully considered their suggestions and made the necessary revisions to enhance the clarity and accuracy of the manuscript. Below, we provide a detailed response to each of their comments.*

*For your convenience, we have marked the reviewer comments in black, our responses in blue and parts of the text reflecting changes in the manuscript in purple with additions/deletions in red. Please note that the line numbers in our responses correspond to the manuscript with tracked changes.*

*Thank you once again for the feedback.*

**Comments from Reviewer 1:**
This is a very well-written and timely study on factors affecting the sensitivity of CI-MS measurements. I warmly recommend its publication, and have only two very small suggestions for further improvement.

We thank the reviewer for the very positive assessment of our manuscript.

1.  As illustrated by reaction R6, the neutral benzene concentration (or neutral to ionised benzene ratio) can affect the detection sensitivity. The same feature applies also to nitrate ionisation (where sensitivities for some compounds can be very different in NO3⁻ - dominated charging regimes vs $(HNO_3)NO_3^-$ dominated ones, indeed some compounds can only be detected in the former). I understand this issue is very specific to the particular combination of charger ion an analyte, and thus somewhat beyond the scope of the present study on general factors affecting CI-MS sensitivity, but maybe you could still briefly discuss it, and how it might be affected by the parameters already studied here?

    Thank you for the comment. We have made the following additions.

    Lines 274-279
    Nitrate ions are very selective and have been routinely used to measure highly functionalized species, such as oxygenated volatile organic compounds (OVOCs) and highly oxidized molecules (HOMs) (Garmash et al., 2024, Alage et al., 2024). At relatively high neutral concentrations of nitric acid in an atmospheric pressure chemical ionization using nitrate ions, most of the reagent ions are clustered with at least one nitric acid molecule. At low neutral nitric acid concentrations, the cluster distribution shifts towards more bare $NO_3^-$ ions, which decreases the selectivity of the ionization scheme. This is because at lower neutral nitric acid

concentrations, weakly interacting analytes do not have to compete as much with neutral nitric acid to form an adduct with $NO_3^-$ (Hyttinen et al., 2015). In benzene cation mode, at extreme differences in concentration, similar effects have been reported (Lavi et al., 2018).

Lines 346/349

Additional secondary effects like changes in the water cluster distribution or the ratio of neutral reagent precursor to charged reagent ions  could also affect the reagent cluster distributions and, in some cases, affect sensitivity distributions, primarily for weakly bound adducts or compounds whose ionization energy is closer to that of the reagent ion.

2. The sentence on line 134 beginning with "Through quantitative..." seems to be missing some words, at least I don't understand the syntax. Please rephrase.

Thanks. The sentence was confusing indeed. Below is the rephrased version.

Lines 134-136

We  assess the statistical variability in normalized instrument performance through quantitative comparison across multiple instruments to gain insight into what parameters govern sensitivity, and therefore control accuracy and precision.

**Comments from Reviewer 2:**
This is a nice manuscript that is leveraging access to many CIMS instruments to probe the drivers of sensitivity. While many studies previous to this one have characterized the sensitivity drivers considered here, the contribution of this work is to show that among multiple instruments we can reproducibly determine sensitivity by holding the controlling factors constant across instruments. While not a surprising result, this is the first work to methodically show that result in quantitative detail. Additionally the authors provide a rather new perspective, that regardless of the ion used the collisional limit should remain constant as it is controlled by geometry, mixing and reaction timescale. I am in support of publication of this work with minimal corrections.

We thank the reviewer for the positive assessment of our manuscript.

1. I feel like this work would benefit from a better discussion on the external sampling factors that would influence net sensitivity. There is minimal discussion of complicating factors like the impact of RH changes when sampling, sampling transmission impact that would go into a net sensitivity function, and potential changes in sampling pressures. The manuscript presents the most stable/idealized version of the sensitivity experiments which may give the reader false confidence that these system do not require frequent calibration checks to ensure stability and correct for transient effects, like RH changes. These types of effects will not impact all ion chemistries equally and individual practices such as material use, operating pressures, and other customizations which are common place among the scientific user community will significantly impact the sensitivity. This feeds into the overall tone of the manuscript where one of the final statements is "... reduce calibration requirements" (page 26, line 724). I do not actually believe any of the work in this manuscript reduces the need for calibration it just provides more detail on the drivers of sensitivity and a better understanding of why comparison of the different customized CIMS instruments over the years have proven challenging. The sensitivity function drivers from real world sampling challenges can not be ignored and will require calibration checks.

   We thank the reviewer for the comment. We agree that instrument-external factors play a relevant role when measuring atmospheric components and assessing overall sensitivity of a certain technique, this is not necessarily specific to CIMS or other mass spectrometric techniques. To make this more clear however we have added the following sentences to the conclusions:

   Lines 723-727

   We note that in this manuscript, we focus on instrument sensitivity, however, the measurement setup, including sampling line material, residence time, and relative humidity and temperature changes of the sample, can influence the net sensitivity function and should be carefully assessed for the desired measurement goals (Neuman et al, 1999; Kürten et al., 2012; Lee et al., 2014; Breitenlechner et al., 2017; Krechmer et al., 2018; Riva et al., 2019; Li et al., 2019).

   Lines 744-746

   These advances in understanding flow tube sensitivity distributions enable better synchronization of sensitivity across instruments,  provide a framework for sensitivity distributions simplifying calibration requirements, and improve measurement comparability between different research groups.

2. Page 3, line 68: The statement "As the number of chemical present in the atmosphere continues to increase" warrants a citation. While I do believe that our awareness of the number of chemicals is increasing due to rapid advancements in measurement technology, I am not sure it is a true statement that the actual number of chemicals is increasing.

Thank you for your comment. You are correct that advancements in measurement techniques have enabled us to detect a growing number of compounds. However, it is important to note that the production of chemicals from human activities is also increasing. For instance, industries are developing new chemicals as legacy compounds face stricter regulations. However, there are no reports of novel chemical species due to the changing climate. We have made the relevant changes in the manuscript as shown below.

Lines 68-70

As the number of chemicals present in the atmosphere continues to increase due to anthropogenic influence (Gibson et al., 2023 and Wang et al., 2024) , the number of possible fragments and interferences becomes ever more challenging to deconvolve by mass spectrometry alone.

3. Page 5, line 134: The sentence that begins "Through quantitative comparison..." needs editing.

   Thanks. The sentence was confusing indeed. Below is the rephrased version.

   Lines 134-136

   We  assess the statistical variability in normalized instrument performance through quantitative comparison across multiple instruments to gain insight into what parameters govern sensitivity, and therefore control accuracy and precision.

4. Page 11, line 329: In the phrase "The sensitivity increases approximately quadratically due to changes in collision frequency.." quadratically in what parameter space? It is unclear what is meant here.

   Thanks for the comment. We have changed the text as follows.

   Lines 335-337

   The sensitivity increases approximately quadratically as a function of reactor pressure due to changes in collision frequency and a proportionally equal increase in reaction time (residence time) at constant mass flow rate through the reactor.

5. Page 14, line 406, "A change in sample gas temperature change..." Delete the second instance of change.

   Done, as suggested.

   Lines 415-416

   In a relative sense, a change in sample gas temperature  from 25 °C to ~100 °C is roughly equivalent to a change of about 20 °C in the reactor temperature.

6. Page 14, line 409. With regards to operating at the lowest feasible temperature, this is specifically true for sensitivity but there may be detrimental impacts from surface absorption and time responses. I think a qualifying statement should be made here.

A statement was added as following:

Lines 419-421

In practice, operating the reactor and inlet at the lowest feasible temperature supports adduct formation while having minimal impact on hydrocarbon detection. However,  the controlled temperature should be sufficiently high enough to prevent surface adsorption or smearing of the analytes of interest, and remain above typical environmental  variations in the vicinity of the reactor and inlet.

7. Page 15, line 436. The discussion on pressure impacts. It is worth noting that there can also be negative impacts from secondary ion chemistry as has been shown by several CIMS studies. Those secondary reactions can complicate interpretation of mass spectra.

We thank the reviewer for pointing this out and have modified the indicated text as following:

Lines 448-450

Increasing reactor pressure generally increases sensitivity due to an increase in collision frequency and reaction time, but can have some penalizing side effects. Higher pressure promotes the formation of higher–order water clusters at a given humidity, which exacerbates water vapour effects and accelerates reagent ion titration, thereby reducing the upper limit of the linear range. Additionally, higher pressure may facilitate enough time for secondary ion chemistry, thereby complicating ionization mechanisms (Zhang and Zhang, 2021; Breitenlechner et al., 2022; Robinson et al., 2022).

8. Page 18, line 513. Your state "we determined that levoglucosan systematically reacts ~20% below collisional limit" but then at several locations through the remainder of the manuscript repeatedly state that "Levo. is known to ionize with iodide anions near the collision limit" (Figure 4 caption), (page 22, line 629), (and elsewhere).

We have rephrased for clarity as below.

Lines 525-530

We selected levoglucosan as the calibrant for the negative ion mode because previous studies have reported that it reacts near the collision limit within the bounds of experimental uncertainty (Lopez-Hilfiker et al., 2016). However, s part of our statistical comparison and investigation into the apparent bias between the collision limit as determined by hydrocarbon calibrations relative to that of levoglucosan (Fig. 4), we determined that levoglucosan is likely to be systematically lower than the collision limit by about ~20 %  with iodide

anions (Fig. S6). This is in contrast to other adduct-forming anions like bromide ions (see Sect. 3.4).

9. Page 22, line 607. You are talking about the N2O5 measurement and state that transmission efficiency changes "... adds to the major challenge of generating a quantitative in situ source of N2O5." However, generating a quantitative calibration source of N2O5 has nothing to do with the instrument transmission. Determining the collisional limit sensitivity of a molecule that has multiple reaction pathways is made difficult when considering the transmission, but having a quantitative N2O5 calibration source and calibrating your peak responses is unrelated to that.

Thanks for the comment. We meant that there are multiple challenges to determining the collision-limited sensitivity using $N_2O_5$. One is that it is difficult to generate a quantitative source in situ, and second, that the reaction of $N_2O_5$ with iodide generates different product ions with very different mass, therefore are more prone to a differing transmission efficiency. We have made the following changes in the text to make it clearer.

Lines 622-624

This dependence, combined with variations in mass transmission efficiency makes it complicated to determine the collision-limited sensitivity. Moreover, it is challenging to generate a quantitative in situ source of $N_2O_5$.